# Lc—ms-based untargeted metabolomics reveals potential mechanisms of histologic chronic inflammation promoting prostate hyperplasia

Jiale Li [1], Beiwen Wu[1], Guorui Fan[1], Jie Huang[1], Zhiguo Li[2]*, Fenghong Cao[1]*

**1** Clinical Medical College, North China University of Science and Technology, Tangshan, China, **2** The Hebei Key Lab for Organ Fibrosis, The Hebei Key Lab for Chronic Disease, School of Public Health, International Science and Technology Cooperation Base of Geriatric Medicine, North China University of Science and Technology, Tangshan, China

* lizhiguo@ncst.edu.cn (ZL); caofenghong@163.com (FC)

## Abstract

### Background

Chronic prostatitis may be a risk factor for developing proliferative changes in the prostate, although the underlying mechanisms are not entirely comprehended.

### Materials and methods

Fifty individual prostate tissues were examined in this study, consisting of 25 patients diagnosed with prostatic hyperplasia combined with histologic chronic inflammation and 25 patients diagnosed with prostatic hyperplasia alone. We employed UPLC-Q-TOF-MS-based untargeted metabolomics using ultra-performance liquid chromatography coupled with quadrupole time-of-flight mass spectrometry to identify differential metabolites that can reveal the mechanisms that underlie the promotion of prostate hyperplasia by histologic chronic inflammation. Selected differential endogenous metabolites were analyzed using bioinformatics and subjected to metabolic pathway studies.

### Results

Nineteen differential metabolites, consisting of nine up-regulated and ten down-regulated, were identified between the two groups of patients. These groups included individuals with combined histologic chronic inflammation and those with prostatic hyperplasia alone. Glycerolipids, glycerophospholipids, and sphingolipids were primarily the components present. Metabolic pathway enrichment was conducted on the identified differentially expressed metabolites. Topological pathway analysis revealed the differential metabolites' predominant involvement in sphingolipid, ether lipid, and glycerophospholipid metabolism. The metabolites involved in sphingolipid metabolism were Sphingosine, Cer (d18:1/24:1), and Phytosphingosine. The metabolites involved in ether lipid metabolism were Glycerophosphocholine and LysoPC (O-18:0/0:0). The metabolites involved in glycerophospholipid

**Data Availability Statement:** All relevant data are within the manuscript and its Supporting Information files.

**Funding:** Update of Financial Disclosure: "This work was supported by the Natural Science Foundation of Hebei Province, China (No. H2019209595). The fund is owned by my advisor, Prof. Fenghong Cao, who is responsible for the conceptualization and project management of the experiment, as well as deciding on the publication of the experiment. No author receives remuneration from the funder. All of our funds were used to carry out the progress of the experiments".

**Competing interests:** The authors have declared that no competing interests exist.

metabolism were LysoPC (P-18:0/0:0) and Glycerophosphocholine. with Impact > 0. 1 and FDR < 0. 05, the most important metabolic pathway was sphingolipid metabolism.

## Conclusions

In conclusion, our findings suggest that patients with prostate hyperplasia and combined histologic chronic inflammation possess distinctive metabolic profiles. These differential metabolites appear to play a significant role in the pathogenesis of histologic chronic inflammation-induced prostate hyperplasia, primarily through the regulation of sphingolipids and glycerophospholipids metabolic pathways. The mechanism by which histologic chronic inflammation promotes prostate hyperplasia was elucidated through the analysis of small molecule metabolites. These findings support the notion that chronic prostatitis may contribute to an increased risk of prostate hyperplasia.

## 1. Introduction

Prostatic hyperplasia (BPH) is a prevalent urologic condition that can adversely influence older men's quality of life. It is also one of the most typical chronic diseases that affect the male population [1]. Pathologic prostate enlargement affects roughly 70% of men aged 61–70 years and 90% of men aged 81–90 years [2]. The Western world is estimated to witness a rise in the number of men aged ≥65 years, and they may constitute up to 20% of the total male population by 2025. Additionally, BPH has progressive characteristics [3]. This condition is likely to burden the healthcare system. Age and androgens are two known risk factors in BPH development. Increased sympathetic nerve activity, metabolic syndrome (MetS), dietary factors, and oxidative stress are additional parameters thought to play a role. However, there is no consensus on which is the main factor. In recent decades, chronic prostatitis has emerged as a potential contributor to the development and progression of BPH [4,5]. However, the mechanism by which it induces BPH remains unclear.

Chronic prostatitis is present in around 43–98% of histological samples of the prostate gland in older men [5,6], providing a rationale to investigate the pathogenesis of BPH. Research indicates that prostate tissue triggers T lymphocytes to generate interferon gamma and IL-17 upon inflammatory stimulation, resulting in the secretion of IL-6 and IL-8 –the primary growth stimulators for prostatic epithelial and stromal cells [7]. Chronic and acute prostatitis can cause oxidative stress in prostatic tissues. Inflammation triggers prostate tissues to generate reactive oxygen species, such as nitric oxide (NO), which creates oxygen free radicals. Macrophages and neutrophils produce a high number of free radicals, leading to oxidative stress on tissues and DNA, ultimately resulting in tissue hyperplasia [8]. In response to oxidative stress, cell membranes produce arachidonic acid. This process is accompanied by the production of oxygen free radicals, which stimulate the release of cyclooxygenase (COX) and contribute to the production of prostaglandins. Prostaglandins play an important role in regulating the proliferation of prostate goblet cells [9]. TGF-b's role has been extensively examined as it is a factor that inflammatory cells secrete. It has been shown to regulate prostatic hyperplasia by stimulating stromal proliferation and differentiation, and is a key factor in androgen control of prostate growth [7]. Chronic prostatitis has been demonstrated in previous studies to be a significant factor in promoting prostate enlargement. However, multiple potential pathogenic mechanisms have been proposed for promoting prostate enlargement by chronic prostatitis, but they remain inconclusive.

Metabolomics is a valuable tool for investigating the pathogenesis of diseases. By conducting qualitative and quantitative analyses of abnormal metabolite changes in the body [10], metabolomics can identify characteristic biomarkers linked to diseases and uncover the potential metabolic pathways of disease development [11]. Ultimately, it finds broad application in diagnosing and researching clinical illnesses. The use of non-targeted metabolomics approaches can detect a diverse range of endogenous metabolites present in body fluids and tissues. These include small molecule metabolites of organic acids, amino acids, fatty acids, sugars, and cholesterol. The three most commonly utilized advanced metabolomics analytical techniques are Nuclear magnetic resonance (NMR), liquid chromatography Mass spectrometer (LC-MS), and Gas chromatography Mass spectrometer (GC-MS). To conduct differential metabolite screening in our study, we employed UPLC-IMS-QToF, which possesses an ion trickling function that can provide information on collision cross-sectional areas and differentiate between isomers [12]. This feature allowed for more precise compound identification, substantially increasing the sensitivity and resolution of the identification process. To date, limited investigations have explored the possible pathogenesis linking chronic prostatitis and the promotion of prostate hyperplasia at the small molecule metabolite level.

In this prospective case-control study, prostate tissues were collected from two groups of patients. One group consisted of patients with prostatic hyperplasia combined with histologic chronic inflammation, and the other group consisted of patients with prostatic hyperplasia alone. The metabolic differences between the two groups of prostate tissues were analyzed using UPLC-IMS-QToF. The mechanism by which chronic inflammation leads to histological prostate hyperplasia was revealed through the analysis of small molecule metabolites. These findings indicate chronic prostatitis as a potential risk factor for prostate hyperplasia and may inform new preventative and therapeutic strategies for prostate enlargement.

## 2. Materials and methods

### 2.1. Patient enrolment and sample collection

The study was conducted in accordance with the guidelines of the Declaration of Helsinki and was approved by the Medical Ethics Committee of the Hospital of North China University of Technology (202103001). All subjects were adults and were provided with written informed consent prior to participation in the study.We enrolled a total of 50 participants from October 1, 2022 to June 1, 2023 from Affiliated Hospital of North China University of Science and Technology located in Tangshan City, Hebei Province- comprising 25 participants who had both prostatic hyperplasia and histologic chronic inflammation, and 25 who had prostatic hyperplasia only. All patients in this study were diagnosed through imaging and evaluated for symptoms of lower urinary tract obstruction, such as urinary frequency, urgency, and slow urine flow. Alternatively, some patients underwent rectal prostate aspiration biopsy for the final diagnosis of prostatic hyperplasia. None of the patients received medication as part of this study. To eliminate any influence from surgical methods on the results, all prostate tissues in this study were obtained by transurethral resection of the prostate (TURP). Based on the postoperative histopathologic findings, we divided participants into two groups: those with BPH combined with histologic chronic inflammation, and those with prostate hyperplasia alone. All prostate tissues were collected within 30 minutes following the surgical procedure and promptly frozen in a refrigerator at -80˚C. Comprehensive clinical and pathological details of the participants are presented in Table 1.

### 2.2. Sample processing for metabolomics

Prostate tissue frozen in a -80˚C freezer was removed and placed on ice to thaw. The thawed prostate tissue was added to pre-cooled methanol and water (1:1, v/v) extraction solvent at a

**Table 1. Clinical and pathological characteristics of the research population.**

| Characteristics | CP group (n = 25) | Control group (n = 25) |
|---|---|---|
| Age(years) | 70.9±6.2 | 69.9±7.5 |
| BMI(kg/m$^2$) | 24.19±3.54 | 23.80±4.06 |
| TG (mmol/L) | 1.35±0.48 | 1.18±0.32 |
| TC (mmol/L) | 4.24±0.80 | 4.19±0.75 |
| PSA (ng/ml) | 7.9±11.2 | 4.3±4.3 |
| Prostate Volume (ml) | 79.68±34.54 | 51.38±18.92 ** |
| IPSS | 26.88±3.40 | 25.64±3.45 |

Table note: ** on behalf of $p < 0.01$, The difference in clinical features of prostatic hyperplasia between the two groups was statistically significant. Abbreviations: TG, total cholesterol; TC, triglycerid; BMI, body mass index; PSA, prostate specific antigen; IPSS, international prostate symptom score; CP, Chronic prostatitis.

ratio of 0.03 mL/mg and mixed on dry ice. The mixture was homogenized with an electric homogenizer for 2 min, and then sonicated for 30 s to break up the tissue thoroughly. After centrifugation (16, 000 g, 4˚C, 10 min), the supernatant and precipitate were collected separately in 2 mL centrifuge tubes. The supernatant was concentrated under vacuum using a Tokyo EYELA CVE-3100 centrifugal concentration unit and dried to powder form (180 min, 45˚C) to obtain the aqueous extract. The precipitate was mixed with pre-cooled dichloromethane/methanol (3:1, v/v) at a ratio of 0.03 mL/mg on dry ice and homogenized for 2 min to thoroughly extract the precipitate. After centrifugation (16, 000 g, 4˚C, 10 min), the precipitate was removed and the supernatant was dried to a powder in a fume hood as an organic extract. Finally, the organic extract was reconstituted in methanol/water (1:1, v/v) at a ratio of 7.2 μL/mg. The mixture was then centrifuged twice (12, 000 g, 4˚C, 10 min) to remove the precipitate and the resulting supernatant was transferred to an injection vial with an internal cannula (Waters, Elstree, UK). The re-dissolution of the aqueous extracts was carried out as above. The aqueous and organic phase extracts were placed on ice and maintained at a temperature between 0 and 4˚C before being transferred to the autosampler and throughout the analysis. To assess instrument stability and data quality, 10 μL of each sample extract was removed and mixed into a Quality control (QC) sample. Periodic injections of QC samples (every 5 samples) were required throughout the analytical run.

## 2.3. UPLC-IMS-QToF-Based tissue metabolomics analysis

Ultra-performance liquid chromatography was used with quadrupole time-of-flight mass spectrometry (UPLC-IMS-QToF) in this experiment. The Waters I-Class Acquity UPLC (Waters, Elstree, UK) was coupled with a Vion IMS QToF (Waters, Elstree, UK). To increase metabolite detection rate, two columns were utilized in this study due to the varying types of metabolites. ACQUITY UPLC ®HSS T3 (1.8 μm, 2.1 × 100 mm; Waters, Ireland) and ACQUITY UPLC ®BEH C8 (1.7 μm, 2.1 × 100 mm; Waters, Ireland) were used for analysis. The aqueous extracts were analyzed using the HSS T3 column (2.1x100 mm, 1.8 μm; Waters, Ireland). The mobile phase A comprised of 0.1% formic acid, and the mobile phase B was a methanol solution with 0.1% formic acid. Gradients were set at 10–0.1% A and 90–99.9% B for 12–21 min, and 0.1–99.9% A and 99.9–0.1% B for 23–24 min. BEH C8 columns (2.1 × 100 mm, 1.7 μm; Waters, Ireland) were utilized for the analysis of organic phase extracts. Mobile phase A was comprised of 0.1% formic acid, while mobile phase B was a combination of methanol and isopropanol in a ratio of 85:15 (v/v) and 0.1% formic acid. Gradients were established at 10–0.1%

A and 90–99.9% B for 17–29 min and 0.1–25% A and 99.9–75% B for 32–33 min. The flow rate of both chromatographic columns was set to 0.4 mL/min at 50°C, with an injection volume of 5 μL, and the temperature maintained at 4°C. Technical abbreviations were explained at first use. Gradient elution was performed following the protocol described by Elizabeth et al [13]. Before the analysis, the system underwent a self-test to reach equilibrium.

The Q/TOF mass spectrometer was used for $MS^E$ scanning in positive and negative ion modes, and two collision energies, high energy and low energy, were used to separate the compounds. The mass spectrometer parameters were set according to the guidelines, with capillary voltages of 2.4 kV (ESI-) and 3.2 kV (ESI+), a sample cone bore voltage of 42 V, a scanning range of m/z 50–1 000 and an ion source temperature of 120°C. The MSE was operated in the positive and negative ion mode with a desolvent gas flow rate of 900L. The desolvent gas flow rate was 900 L/h at a desolvent gas temperature of 350°C, and the cone-well gas flow rate was 25 L/h. Leucine enkephalin was used for real-time correction of the mass axis.

## 2.4. Data processing and analysis

We acquired raw data with the Waters UNIFI 1.8.1 Workstation UPLC-Q-TOF/MS Acquisition software and processed the mass spectrometry data with Progenesis QI 3.0 qualitative and quantitative software (Waters, Nonlinear Dynamics, Newcastle, UK). A metabolomics data processing workflow was established, consisting of various processes, such as baseline filtering, peak identification, integration, retention time correction, peak alignment, and normalization. Finally, we exported the normalized data matrix from QI and imported it into the SIMCA software package (Umetrics AB, Sweden). To observe the overall distribution among the samples and ensure the stability of the entire analytical process [14], we initially used unsupervised principal components analysis (PCA). Then, to identify the overall differences in metabolic profiles among the groups [15], we used supervised orthogonal partial least squares discriminant analysis (OPLS-DA). Multidimensional and unidimensional analyses are often combined to screen for differential metabolites between groups. In OPLS-DA analysis [16], variable weight values (VIP) can be utilized to determine the strength of influence and explanatory power of each metabolite's expression pattern in relation to the categorical categories of the sample groups. The criteria for screening were VIP > 1, Anova $p < 0.05$ and Fold change (FC)> 1.2 or < 5/6, which ensured the significance and importance of different metabolites between groups. Metabolites were identified and annotated using the ChemSpider plug-in in QI, which combines the Human Metabolome Database (HMDB), Kyoto Encyclopedia of Genes and Genomes (KEGG) databases. The main parameters for database identification were set as follows: mass tolerance of 5 ppm, fragment mass tolerance of 5 ppm, isotopic similarity >80%, and predicted collision cross section (CCS) tolerance of ±5%. Pathway enrichment analysis and heat mapping of differential metabolites were performed using MetaboAnalyst (https://www.metaboanalyst.ca/). Visualizing the connections between samples and differences in metabolite expression can enhance understanding of the mechanisms driving changes in metabolic pathways among differential samples. The diagnostic performance of differential metabolites can be assessed by plotting subject operating characteristic (ROC) curves in GraphPad Prism version 8.0.1 and measuring the area under the curve (AUC) for diagnostic accuracy. Additionally, GraphPad Prism version 8.0.1 was used to construct bar graphs and pie charts.

The participants' pathological and clinical information were statistically described as mean ± standard deviation (SD). Statistical significance was determined by employing Student's t-test, with a significance level of P < 0.05. The statistical analysis was performed using SPSS 26.0 software.

## 3. Results

### 3.1. Inflammatory infiltration plays a role in the development of BPH

We first compiled the clinical and pathological data of participants and conducted statistical analysis. Analysis of Table 1 reveals that there were no significant differences between the two groups in terms of age, Body Mass Index (BMI), total cholesterol (TC), triglyceride (TG), prostate-specific antigen (PSA), and international prostate symptom score (IPSS), thereby minimizing interference of multiple factors on experimental outcomes. However, the difference in prostate volume (PV) (79.68 vs. 51.38, P < 0.01) was statistically significant. To better illustrate the difference in prostate volume between patients with prostatic hyperplasia combined with histological chronic inflammation and those with prostatic hyperplasia alone, we used bar charts to display the statistical data we obtained. As illustrated in Fig 1, the prostate volume in patients with prostatic hyperplasia combined with histologic chronic inflammation was significantly greater compared to patients with prostatic hyperplasia alone. This finding indicates that inflammatory infiltration is a significant factor in the onset and progression of BPH.

### 3.2. Differential prostatic metabolites between BPH combined with histological chronic inflammation and BPH patients

Metabolic profiles of prostate tissue specimens from 25 patients with prostatic hyperplasia combined with histological chronic inflammation and 25 patients with prostatic hyperplasia

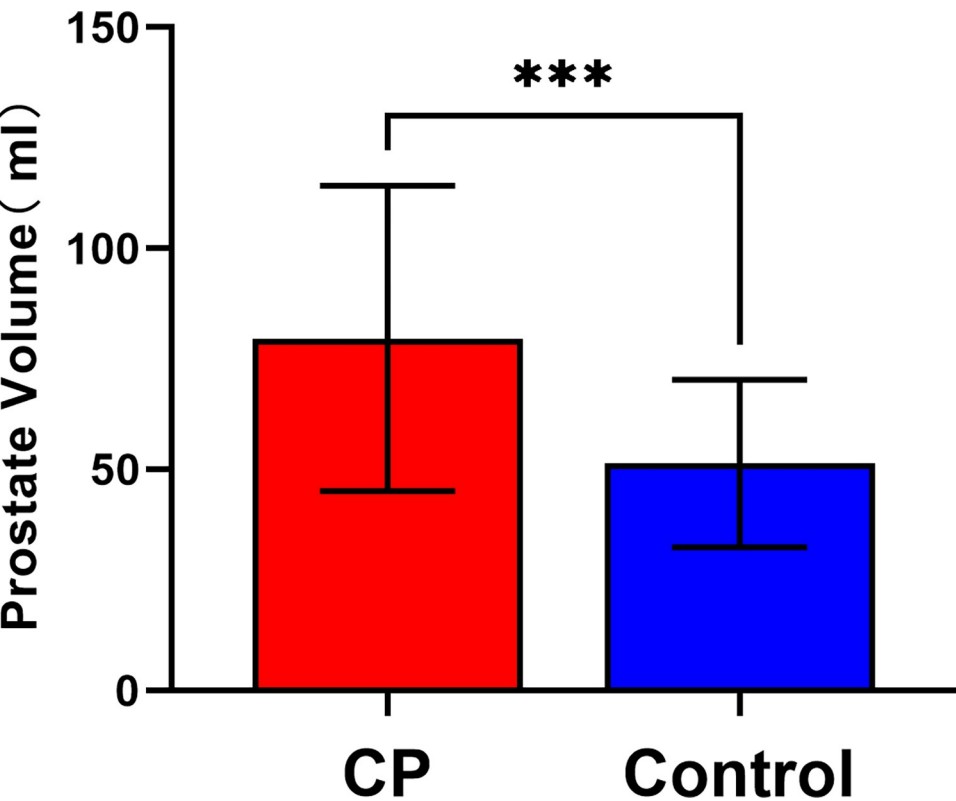

**Fig 1. The prostate volume in patients with benign prostatic hyperplasia combined with histological chronic inflammation is significantly larger than that in patients with simple benign prostatic hyperplasia.** Data are presented as means ± SD, *** *P* < 0.001.

alone were analyzed using an untargeted metabolomics approach based on ultra-high-performance liquid chromatography-high-resolution mass spectrometry. 40,463 and 26,883 features were separated using HSS T3 and BEH C8 columns, respectively. To investigate the variation of metabolic patterns between them, the unsupervised model PCA was used in this study (Fig 2A, 2C, 2E and 2G). The PCA score plots depict clear clustering within the 95% confidence interval for both the positive and negative ion models with minimal outliers, indicating reliable data. The QC samples were also clustered, suggesting instrument stability and data confidence. Subsequently, the supervised discriminant statistics method OPLS-DA was used (Fig 2B, 2D, 2F and 2H). Both groups were well separated in the positive and negative ion modes, suggesting significant changes in the metabolic profile of prostate tissue in the group of patients with prostatic hyperplasia combined with histologic chronic inflammation and identifying important metabolites contributing to metabolic differentiation. A total of 19 important differential metabolites were screened out using the important metabolites with VIP>1 obtained from the OPLS-DA model, combined with Anova $p<0.05$ and FC>1.2 or <5/6 in the normalized data matrix (Table 2). The main components included glycerolipids, glycerophospholipids and sphingolipids. In order to more visually show the differences of metabolites between the two groups, heat maps (Fig 3) were plotted for the identified differential metabolites. The heat map (Fig 3A) demonstrated the differences in the expression of the differential metabolites between the individual samples. The heat map (Fig 3B) more vividly showed the average distribution of differential metabolites between the two groups. Nine metabolites were upregulated and 10 metabolites were downregulated in the metabolic profiles of prostate tissues from patients with prostate hyperplasia combined with histologic chronic inflammation. All metabolites were categorized according to superclasses (Fig 4A), with 84% of metabolites being lipids and lipid-like molecules (including 44% glycerophospholipids, 25% glycerolipids, 6% sphingolipids, and 25% others) (Fig 4B) and 11% organic nitrogen compounds (50% sphingosine and 50% phytosphingosine) (Fig 4C).

### 3.3. Evaluation of diagnostic value of differential metabolites

To determine the diagnostic efficacy of the differential metabolites, the obtained differential metabolites were analyzed by subject operating characteristic curve (ROC). Six metabolites

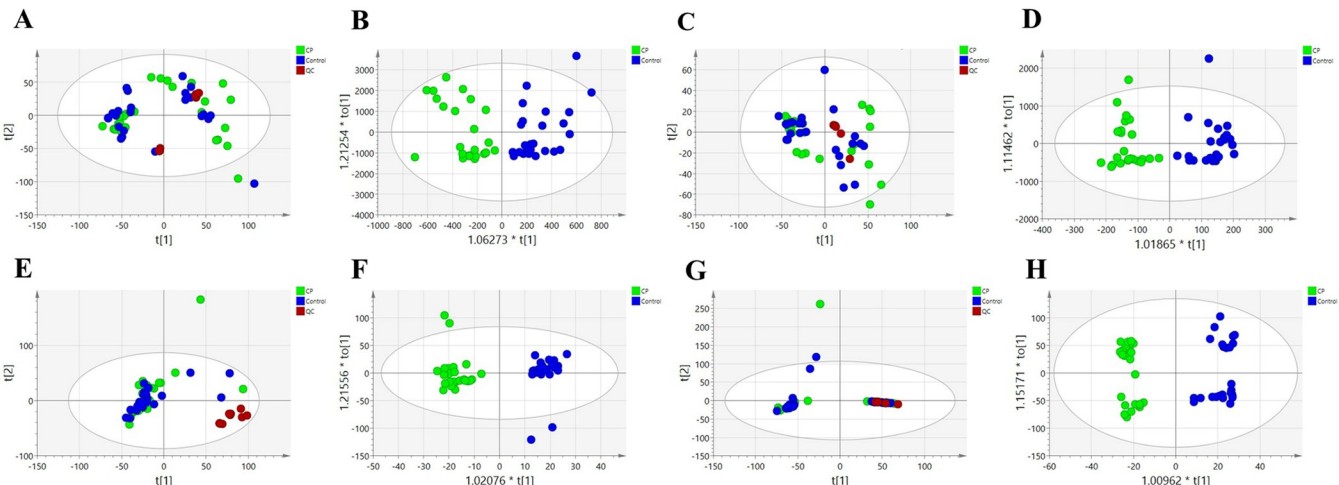

**Fig 2. PCA and OPLS-DA of aqueous extracts and organic matter.** Multivariate statistical analysis of organic phase extracts, ESI (+), **(A)** PCA score plot, **(B)** OPLS-DA score plot; ESI (-), **(C)** PCA score plot, **(D)** OPLS-DA score plot. Multivariate statistical analysis of aqueous extracts, ESI (+), **(E)** PCA score plot, **(F)** OPLS-DA score plot; ESI (-), **(G)** PCA score plot, **(H)** OPLS-DA score plot.

**Table 2. Differential metabolites for distinction of CP and control group.**

| HMBD ID | name | Measured m/z | RT (MIN) | Adducts | Measured CCS (Å2) | Predicted CCS (Å2) | Anova (p) | Fold Change | VIP | Trend |
|---|---|---|---|---|---|---|---|---|---|---|
| HMDB0112120 | FAHFA(18:0/12-O-18:0) | 584.56 | 4.44 | M+NH4 | 276.4 | 266.9 | 0.026 | 1.56 | 4.42 | ↑ |
| HMDB0030965 | 9-Octadecenal | 284.29 | 4.00 | M+NH4 | 184.3 | 179.1 | 0.022 | 25.33 | 1.03 | ↑ |
| HMDB0002356 | Hexacosanoic acid | 414.42 | 3.55 | M+NH4 | 225.8 | 223.8 | 0.011 | 2.36 | 2.15 | ↑ |
| HMDB0007071 | DG(15:0/18:0/0:0) | 600.55 | 3.53 | M+NH4 | 278.3 | 265.2 | 0.027 | 1.40 | 1.46 | ↑ |
| HMDB0000086 | Glycerophosphocholine | 258.10 | 0.57 | M+H | 155.6 | 155.6 | 0.012 | 2.30 | 1.06 | ↑ |
| HMDB0061690 | LysoPE(18:1(11Z)/0:0) | 619.28 | 1.64 | M-H | 244.4 | 244.7 | 0.040 | 4.13 | 8.68 | ↑ |
| HMDB0010581 | PG(16:0/22:4(7Z,10Z,13Z,16Z)) | 797.53 | 9.25 | M-H | 296.1 | 283.6 | 0.044 | 6.06 | 1.81 | ↓ |
| HMDB0243890 | 1-O-Hexadecyl-sn-glycero-3-phosphocholine | 482.35 | 12.69 | M+H | 237.8 | 231.7 | 0.017 | 1.45 | 3.17 | ↓ |
| HMDB0011143 | MG(O-18:0/0:0/0:0) | 367.31 | 14.51 | M+Na | 205.1 | 201.7 | 0.025 | 2.67 | 1.76 | ↓ |
| HMDB0011149 | LysoPC(O-18:0/0:0) | 510.39 | 13.47 | M+H | 248.1 | 240.1 | 0.011 | 1.83 | 3.05 | ↓ |
| HMDB0013122 | LysoPC(P-18:0/0:0) | 508.37 | 13.33 | M+H | 241.5 | 237.3 | 0.029 | 1.52 | 1.77 | ↓ |
| HMDB0062305 | 1-(11Z-eicosenoyl)-glycero-3-phosphate | 487.27 | 13.07 | M+Na | 224.5 | 222.5 | 0.046 | 1.65 | 2.17 | ↓ |
| HMDB0038229 | Bn-NCC-1 | 734.30 | 5.89 | M+NH4 | 256.6 | 263.6 | 0.039 | 1.51 | 1.32 | ↓ |
| HMDB0011553 | MG(0:0/22:2(13Z,16Z)/0:0) | 428.37 | 11.82 | M+NH4 | 226.3 | 215.2 | 0.031 | 1.30 | 1.44 | ↓ |
| HMDB0072856 | MG(17:0/0:0/0:0) | 362.32 | 10.17 | M+NH4 | 203.2 | 198.4 | 0.031 | 1.33 | 7.25 | ↑ |
| HMDB0000252 | Sphingosine | 300.28 | 10.16 | M+H | 193.5 | 185.8 | 0.015 | 1.99 | 1.01 | ↑ |
| HMDB0004610 | Phytosphingosine | 318.29 | 10.14 | M+H | 194.7 | 194.0 | 0.033 | 1.33 | 10.38 | ↑ |
| HMDB0004953 | Cer(d18:1/24:1) | 670.60 | 22.19 | M+Na | 288.5 | 278.5 | 0.017 | 1.53 | 3.18 | ↓ |
| HMDB31066 | 16-Methylheptadecanoic acid | 283.26 | 14.47 | M-H | 183.2 | 180.2 | 0.011 | 1.70 | 1.62 | ↓ |

Table note: Trend represent the rise or fall of differential metabolites in the CP group.

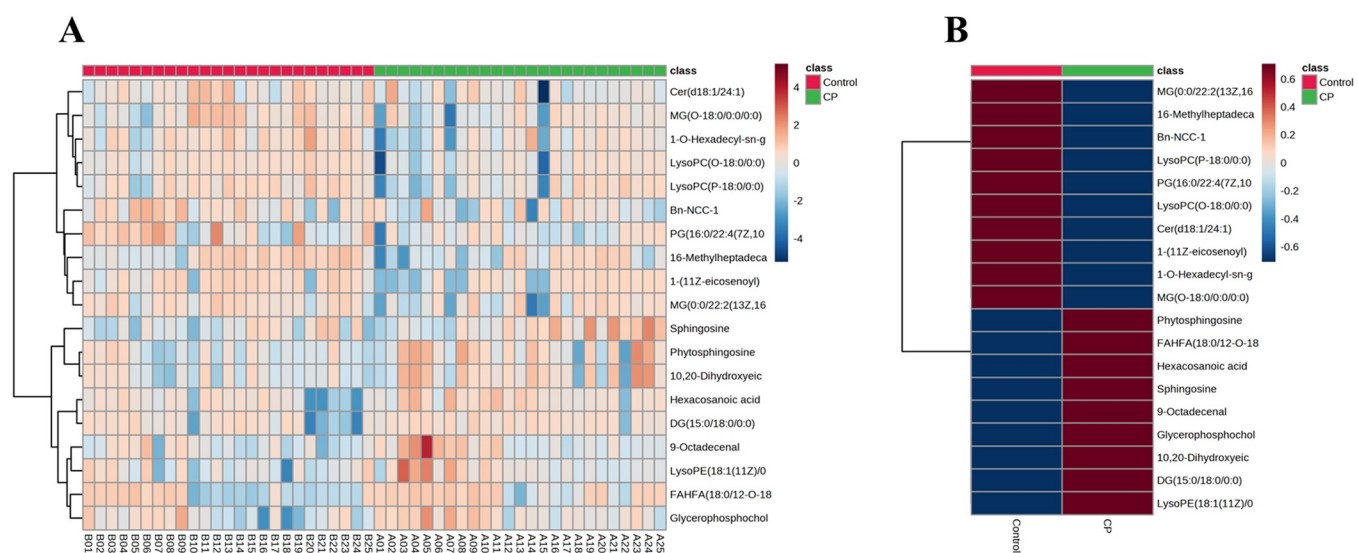

**Fig 3. Differential metabolite heat maps in modes.** The columns represent samples, the rows represent metabolites, and the relative content of the metabolites is displayed by color. **(A)** the differential metabolites between the CP group and the control group, **(B)** the average distribution of differential metabolites between the two groups.

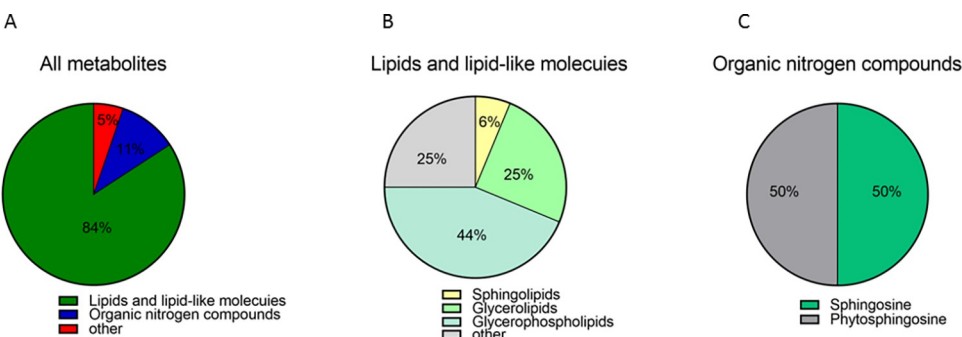

**Fig 4. Classification of metabolites. (A)** The superclass of all metabolites; **(B)** Classification of lipids and lipid-like molecules; **(C)** Classification of Organic nitrogen compounds.

with an AUC area greater than 0. 7 were considered to be metabolites with better diagnostic efficacy. The diagnostic efficacy of each metabolite is shown, including Phytosphingosine, 1-O-Hexadecyl-sn-glycero-3-phosphocholine, LysoPC (O-18:0/0:0), 1-(11Z-eicosenoyl)-glycero-3-phosphate, MG (17:0/0:0/0:0), and 16-Methylheptadecanoic acid. In order to improve the identification of the disease, we constructed a prediction model by combining the six metabolites. Eventually, we also obtained an optimal prediction model with an AUC area of 0.7792. As shown (Fig 5A–5G), we visualized the diagnostic efficacy of the individual metabolites and their combined models in an ROC plot.

## 3.4. Metabolic pathway analysis

The 19 differential metabolites screened were imported into MetaboAnalyst 5. 0 for metabolic pathway enrichment analysis (Fig 6). The analysis results showed that there were three metabolic pathways, including sphingolipid metabolism, ether lipid metabolism and glycerophospholipid metabolism. The most important metabolic pathway among them was sphingolipid metabolism by the criterion of Impact > 0. 1 and satisfying $FDR < 0.05$ (Table 3). Three of the differential metabolites were involved in sphingolipid metabolism, which were Sphingosine, Phytosphingosine and Cer (d18:1/24:1).

## 4. Discussion

BPH is a prevalent chronic progressive disease. Clinical BPH is described as the combination of lower urinary tract symptoms (LUTS) and benign prostatic enlargement. Age and androgens are acknowledged as key factors for BPH development, but the pathogenesis and cellular and molecular mechanisms leading to the disease's symptoms are still not fully comprehended [17]. Recent decades have seen significant progress in research in this field. Basic science and clinical studies suggest that inflammation may be a central mechanism in the histologic development, prostate enlargement, and disease progression of BPH. A study by Kim et al. found chronic inflammation of the prostate in 80% of patients with BPH. It is suggested that prostate inflammation plays an important role in the progression and pathogenesis of BPH [18]. Several studies have established a strong correlation between prostatic hyperplasia and inflammatory patterns, as indicated by a higher prevalence of signs of chronic inflammation (61%) in larger prostates (80–89 mL) compared to smaller ones [5]. Moreover, clinical data indicate that chronic inflammation plays a crucial role in the development of BPH. In this study, prostate tissues were collected from 25 patients diagnosed with prostatic hyperplasia alone and another 25 patients diagnosed with prostatic hyperplasia combined with histologic chronic

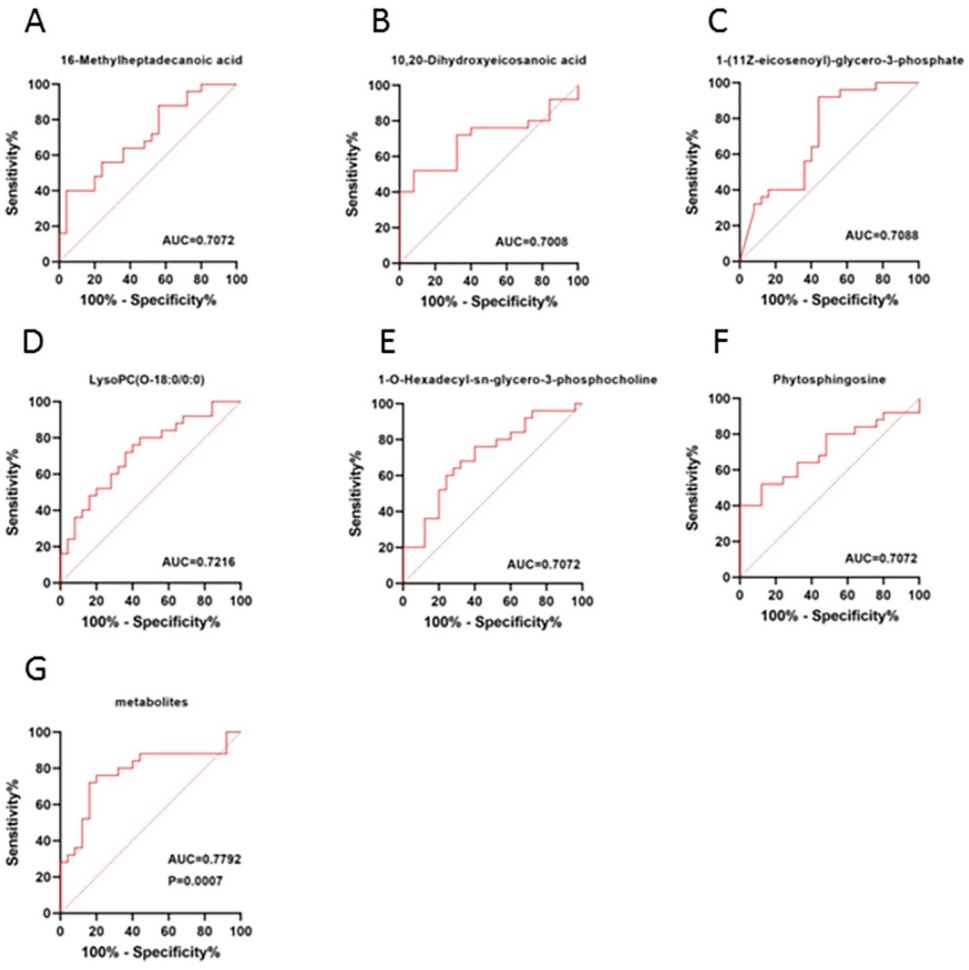

**Fig 5. ROC curve analysis of the selected metabolites for discriminating CP from control groups. (A-F)** The ROC plots are as follows: 16-Methylheptadecanoic acid, 10,20-Dihydroxyeicosanoic acid, 1-(11Z-eicosenoyl)-glycero-3-phosphate, LysoPC(O-18:0/0:0), 1-O-Hexadecyl-sn-glycero-3-phosphocholine, Phytosphingosine, **(G)** a combination of six metabolites.

inflammation. Based on a statistical analysis of clinical characteristics, it was found that patients with prostatic hyperplasia combined with histologic chronic inflammation had a larger prostate volume compared to the control group. This discovery aligns with previous research findings. Inflammatory response occurs in the body as a result to harmful stimuli. Its activation leads to multiple molecular mechanisms, and both local and systemic reactions. Such responses indicate systemic metabolic imbalances within affected tissues [19]. Upon reviewing relevant literature, there is a limited number of metabolomic studies conducted to investigate the role of chronic prostatitis in relation to BPH development. Therefore, the current investigation employed untargeted metabolomics to elucidate the possible pathways through which histologic chronic inflammation stimulates prostate hyperplasia.

A non-targeted high-throughput liquid chromatography-mass spectrometry was used to perform a comprehensive metabolomic analysis of prostate tissues from patients with prostatic hyperplasia and histological chronic inflammation. The aim was to investigate the potential mechanisms by which histological chronic inflammation promotes prostatic hyperplasia. Before, nineteen metabolite levels differed significantly between patients with prostatic hyperplasia combined with histological chronic inflammation and those with prostatic hyperplasia

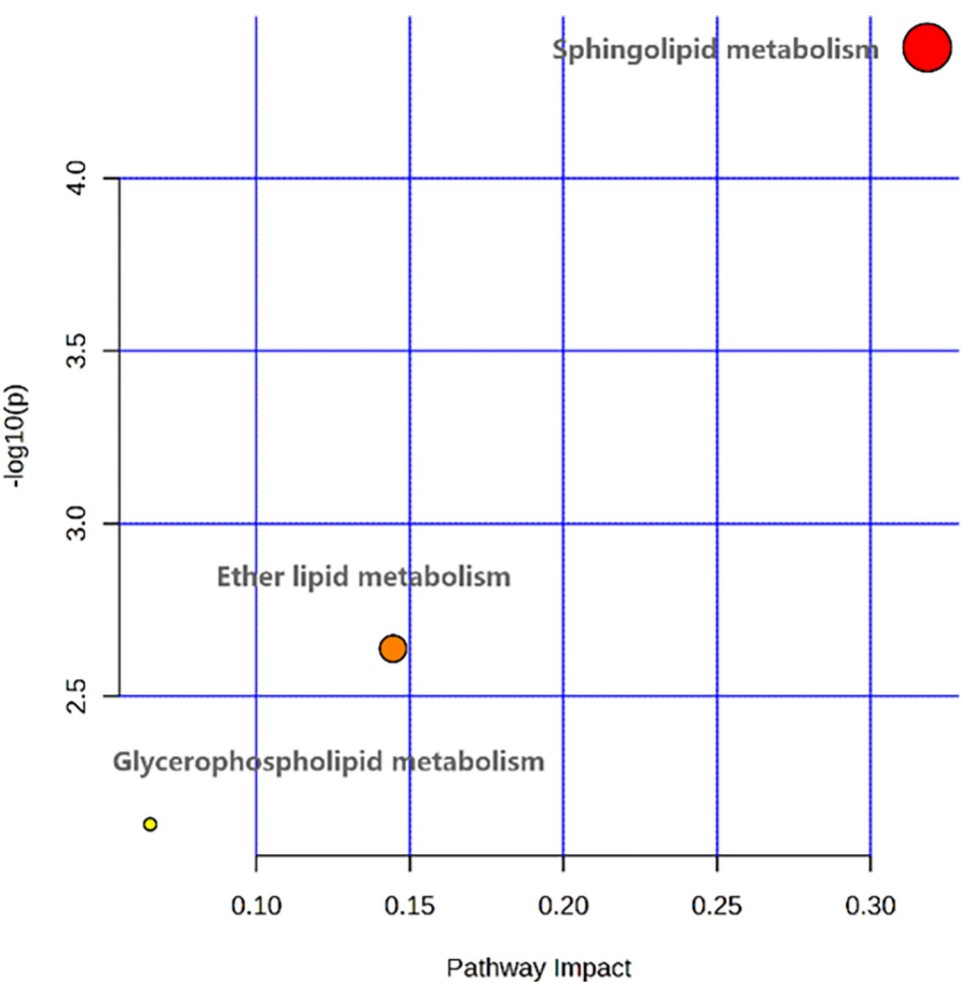

**Fig 6. Significantly altered metabolic pathways and metabolites between CP and control groups.** Pathway enrichment analysis from absolutely identified metabolites in (A) KEGG databases. Graph shows the impact the statistically significant identified metabolites have in the different pathways (x-axis) together with the statistical significance of those metabolites (y-axis). Highlighted pathways have several statistically significant metabolites that in turn produce a high impact in the pathway.

alone. Eighty-four percent of the identified metabolites were lipids and lipid-like molecules, including glycerolipids, glycerophospholipids, and sphingolipids. Our research uncovered that six of these differential metabolites had a strong predictive potential for histologic chronic inflammation, which contributes to prostate hyperplasia. The highest predictive ability was

**Table 3. Different metabolic enrichment pathways between CP and control groups.**

| Pathway ID | Pathway Name | Match Status | Raw *P* | FDR | Impact |
|---|---|---|---|---|---|
| hsa00600 | Sphingolipid metabolism | 3/21 | 4.1827E-5 | 0.0035135 | 0.31846 |
| hsa00565 | Ether lipid metabolism | 2/20 | 0.0023014 | 0.096657 | 0.14458 |
| hsa00564 | Glycerophospholipid metabolism | 2/36 | 0.0074218 | 0.20781 | 0.0655 |

Table note: Pathway ID: KEGG pathways; Match status: Metabolites involved in the pathway. Data before "/" indicates the number of metabolites currently involved in the pathway; The number after "/" indicates the total number of metabolites in the current pathway; Impact value: Represents the overall importance score of the pathway, with a total score of 1, which can be calculated based on the relative position of metabolites in the pathway.

demonstrated by LysoPC (O-18:0/0:0). The predictive model revealed better diagnostic efficacy when the six differential metabolites were combined. Metabolic pathway analysis indicates a potential association between histologic chronic inflammation promoting prostate hyperplasia and complications in sphingolipid metabolism, ether lipid metabolism, and glycerophospholipid metabolic pathways. The most important metabolic pathway was sphingolipid metabolism, as defined by Impact $> 0.1$ and meeting $FDR < 0.05$.

Lipid metabolism is a critical metabolic process with a substantial impact on maintaining the body's internal environment's homeostasis. Lipids play a crucial role in regulating various life activities, including energy conversion, structural support, substance transport, cell development and differentiation, and apoptosis. Moreover, they serve as pivotal first and second messengers for cells and participate in numerous physiological processes of organisms via lipid-lipid interactions and interactions between lipids and other biomolecules, creating an intricate lipid metabolic network [20–22]. Previous research has demonstrated that inflammation can interrupt the expression of specific molecules involved in lipid metabolism within tissue cells, resulting in lipid transport and distribution issues throughout the body. These disorders can ultimately lead to damage to tissues and organs [23–25]. The findings from large-scale epidemiological studies conducted abroad demonstrate that individuals with chronic inflammatory illnesses, including systemic lupus erythematosus and rheumatoid arthritis, exhibit apparent impairments in lipid metabolism [26,27]. This implies that enduring inflammatory stimulation may be a key contributor to the development of lipid metabolism disorders. Elevated free fatty acids in the blood indicate early stages of lipid metabolism disorders. Technical term abbreviations are explained when first used. Studies suggest that triglyceride levels increase at the beginning of the infectious inflammatory response. This is associated with the augmented lipolysis of adipose tissue that is caused by inflammatory factors, particularly tumor necrosis factor. The outcome is an augmented release of triglycerides from very low-density lipoproteins and increased hepatic synthesis of fatty acids [28]. Chronic inflammation results in heightened consumption of oxygen at the tissue level, leading to severe hypoxia, hastened breakdown of adipose tissue, and accumulation of fatty acids. This study identified that 25% of the distinct metabolites were fatty acyls, out of which 75% exhibited high levels in the chronic prostatitis (CP) group, which aligns with prior research. When the body requires energy, fatty acyl groups break down into fatty acids and glycerol. The fatty acids are then oxidized to produce energy reactions, which in turn source energy for cell proliferation within the body. Furthermore, they serve as vital signaling molecules. Recent studies have identified an orphan G protein-coupled membrane receptor family of Free Fatty Acid (FFA) in the cell membrane. This receptor family mediates the free fatty acid through the ERK, PI3K-Akt, and MAPK signaling pathways. Its significant role in regulating insulin hormone secretion [29,30], lipid metabolism [31], cell proliferation and differentiation [32], apoptosis [33], and immune response [34] is widely recognized. Long-chain fatty acids (LCFA) play a significant regulatory role in insulin hormone secretion, lipid metabolism, cell proliferation, differentiation, apoptosis, and immune response. The binding of LCFA to the GPR120 receptor activates ERK and PI3K-Akt, eliciting a series of downstream responses [33]. In STC-1 cells, it induces insulin release while promoting cell proliferation and inhibiting apoptosis via the PI-3K/Akt signaling pathway. Furthermore, HEK-293 cells which are derived from human embryonic kidney cells and overexpress GPR120 showed the ability to regulate cell growth as well as inhibit programmed cell death through the PLC-MAPK pathway in response to LCFA. Therefore, the buildup of free fatty acids may have a crucial impact on the progression of prostate hyperplasia induced by chronic inflammation.

Sphingolipids (SP) play a crucial role in intracellular homeostasis and have strong links to the development of various diseases [35]. These molecules, which are both hydrophobic and

hydrophilic, are essential components of the plasma membrane in nearly all vertebrate cells, playing crucial roles in regulating a wide range of cellular functions [36]. Among these essential functions are cell-to-cell interactions, cell adhesion, cell proliferation, cell migration, and cell death, all of which are regulated in part by sphingolipids [36]. The essential elements maintaining sphingolipid metabolism homeostasis include sphingolipid intermediates, ceramides, and sphingosine-1-phosphate (S1P), which exert contrasting roles in cell longevity and development [37]. Thus, controlling the equilibrium among them is crucial in apoptosis, differentiation, and proliferation. Ceramides are known triggers of apoptosis [38] and impact events such as IR, oxidative stress (OS), inflammation, and apoptosis [39,40]. Ceramide synthesis includes three pathways [41,42], with one of them being sphingomyelinase-catalyzed SM hydrolysis [43]. Previous studies have shown that sphingolipids regulate cell signaling pathways in inflammation [44–46]. In brief, inflammatory cytokines such as TNFα and IL-6 activate sphingomyelinase, leading to the hydrolysis of sphingomyelin into ceramide. Ceramidase then cleaves ceramide to regenerate sphingomyelin. Additionally, prostate inflammation generates free radicals like nitric oxide (NO) and various reactive oxygen species, which can be sourced from macrophages and neutrophils. These free radicals, through oxidative stress on tissues and DNA, can induce proliferative transformation [47]. DNA damage triggers the activation of ATM and p53, which subsequently disrupts sphingolipid metabolism, causing the accumulation of ceramide (Cer) and/or sphingomyelin. Depending on the severity of the DNA damage, p53 promotes the hydrolysis of Cer to sphingomyelin by activating human alkaline ceramidase 2. In cases of severe DNA damage, this leads to the accumulation of sphingomyelin and the induction of apoptosis. Conversely, after mild DNA damage, it results in senescence and cell cycle arrest [48–51]. In the present study, we found that prostate tissue metabolites from patients with histologic inflammation combined with prostate hyperplasia had decreased levels of Cer (d18:1/24:1), Sphingosine and Phytosphingosine levels were increased, which seems to be inconsistent with the findings of previous studies. However, it is well known that prolonged chronic inflammation leads to localized tissue hypoxia, high production of growth factors and inflammatory mediators. Reviewing the literature of others, growth factors (PDGF, IGF, VEGF), cytokines (TNF, IL-1), and hypoxia all activate sphingomyelin kinase, which elevates S1P levels [52]. The "sphingolipid variable resistor" concept proposes that the equilibrium between two key metabolites of sphingolipid metabolism, ceramide (Cer) and sphingosine-1-phosphate (S1P), dictates cell fate. Sphingosine kinases are pivotal in maintaining this balance by catalyzing the production of pro-survival S1P while concurrently degrading pro-apoptotic sphingosine and Cer [46]. It is therefore plausible to hypothesize that under sustained inflammatory stimuli, the balance of sphingolipid transducers shifts towards an increase in S1P, a more soluble sphingolipid that typically exists at lower concentrations within cells but is abundant in serum, associated with lipoproteins and albumin. Extracellular sphingosine-1-phosphate (eS1P) acts as a high-affinity ligand for five G-protein-coupled receptors (S1PR1 to S1PR5), triggering downstream signaling cascades that influence cell migration and survival [53]. S1P can also exert its functions independently of the S1P receptor within the cell, potentially eliciting similar or contrasting downstream effects compared to extracellular S1P (eS1P) signaling. The signaling mediated by S1P receptors has been thoroughly investigated and has been reviewed extensively by Maceyka et al. [54] and Pyne et al. [55] particularly in terms of its important functions in promoting angiogenesis, cell proliferation and endothelial cell migration, proliferation and survival. The present study found altered levels of sphingolipid metabolites in prostate tissues of patients with histologic inflammation combined with prostatic hyperplasia, which may suggest that chronic inflammation promotes prostatic hyperplasia by maintaining a balance between ceramide (Cer) and S1P.

In mammalian cells, glycerophospholipids (GP) make up about 60% of the overall lipid composition, and they are a basic phospholipid that forms the structure of cell membranes [56]. Previous studies have shown that disorders of glycerophospholipid metabolism are associated with inflammation and that phospholipids may act as inflammatory mediators [19,57]. In the present study, we identified a total of 16 lipid and lipid-like molecules, of which glycerophospholipids accounted for 44%. Notably, we derived the involvement of LysoPC (P-18:0/0:0) in the glycerol metabolic pathway by metabolite pathway enrichment analysis. Subject work characteristic curve (ROC) analysis revealed LysoPC (O-18:0/0:0) to have the best diagnostic efficacy. Both lysophosphatidylcholine levels were reduced. The superfamily of phospholipase A2 (PLA2) enzymes is responsible for the hydrolysis of fatty acids in the sn-2 position of membrane phosphatidylcholine (PC), yielding fatty acids that can be metabolized to a variety of proinflammatory eicosanoids as well as to related LPCs [58]. In contrast to our findings, some evidence suggests that phospholipase activity increases under inflammatory conditions [59–61]. However, the decrease in production may be justified by the fact that glycerophosphorylcholine, one of the main precursors of lysoPC, decreases with increasing hsCRP levels [19]. Interestingly, population-based studies have shown that lysoPC 18:1 and 18:2 are negatively associated with the risk of developing T2DM [62] or coronary heart attacks [63]. Given that a chronic inflammatory environment is a hallmark of such diseases, further support for this study that a reduction in lysophosphatidylcholine may be justified. The researchers found that plasma levels of LPC were significantly reduced in a mouse model and in patients with sepsis [64,65], and were lower during bacterial infections. LPC also had anti-infective effects that inhibited excessive inflammation. The anti-inflammatory effect was related to the length and degree of saturation of the acyl chain of LPC [66]. The 18:0 LPC obtained in this study alone reduced the concentration of inflammatory factors such as TNF-α, IL-6, IL-α, IL-1β, and IL-10 [65,67]. Lysophosphatidylcholine (LPC) and its byproduct lysophosphatidic acid (LPA), which are signaling molecules involved in key aspects of cell and tissue biology, such as plasma membrane shaping, cell growth and death, and inflammatory cascade responses. There are many mechanisms by which LPC induces apoptosis, such as cystatinase activation, calcium inward flow, cytochrome C release, and mitochondrial pathways [68,69]. In addition, LPC can induce apoptosis by increasing FasL expression through activation of the NF-κB signaling pathway [70]. In the present study, the reduction of LPC content inhibited multiple apoptotic mechanisms. Previous authors have suggested that the inflammatory process may directly promote prostate enlargement by stimulating prostate growth or by reducing prostate apoptosis. This view coincides with the present results. In vivo LPC can be converted to LPA by the action of ATX, and thus the effects attributed to LPC can be mediated, at least in part, by LPA. A review of others' findings that ATX is highly expressed in nonmalignant tissues such as arthritic synovium, fibrotic lungs, post-traumatic reactive astrocytes [71], frontal cortex of patients with Alzheimer's disease-like dementia, and cerebrospinal fluid of patients with multiple sclerosis [72] suggests that ATX may be involved in chronic inflammatory disease. A rat air sac model established by J.K. Gierse et al. was tested to validate that ATX at sites of inflammation induction may stimulate local LPA production and its auto- and paracrine effects, and the results suggest that ATX is a major source of LPA during inflammation [73]. LPA is a biologically active phospholipid with diverse functions in almost all mammalian cell types. Multiple LPA effector functions can be attributed to at least six G protein-coupled LPA receptors. Each LPA receptor to the corresponding signaling of its associated G protein [74], which in turn controls a large number of cellular functions. gαi stimulates the pro-mitotic Ras-Raf-MEK-ERK pathway, as well as phosphatidylinositol 3-kinase β promotes cell survival and many other cellular functions. LPA has been reported to synergize with PDGF and EGF to promote cell migration and proliferation [75], suggesting

that the ATX/LPA axis may be important in regulating and/or amplifying growth factor responses. In the present study, LPA was not identified as a differential metabolite, possibly due to the fact that the tissue concentration of LPC is much higher than that of LPA. Thus potential mechanisms by which chronic inflammation promotes prostate hyperplasia may be related to the PLA2/LPC and ATX/LPA axes.

However, there are limitations to this study. First, it was conducted at a single location, the North China University of Science and Technology Hospital, and was not a multicenter study. Second, the study cohort was small and there was no external validation cohort, resulting in overfitting bias in the model. Finally, a multi-omics analysis that integrates data on genes, proteins, and metabolites should be conducted to gain deeper insight into the underlying mechanisms of histologic chronic inflammation that promote prostate hyperplasia.

## 5. Conclusion

In conclusion, we found that histological chronic inflammation results in marked changes in prostate metabolic profiles, particularly lipid metabolites, such as free fatty acids, ceramides, sphingomyelins, and lysophosphatidylcholine, using LC-MS-based metabolomics to analyze prostate tissue samples. These metabolites play a role in the pathogenesis of histological chronic inflammation-induced prostate hyperplasia, mainly by controlling SP and GP metabolic pathways. These findings may offer novel perspectives on preventing and treating prostatic hyperplasia.

## Supporting information

**S1 Table. Expression of differential metabolites between CP and control groups.** See S1 Table of the Additional Document for details.
(CSV)

## Acknowledgments

The authors would like to thank all the volunteers and patients for their participation.

## Author Contributions

**Data curation:** Beiwen Wu.

**Formal analysis:** Jie Huang.

**Funding acquisition:** Fenghong Cao.

**Investigation:** Zhiguo Li.

**Supervision:** Guorui Fan.

**Writing – original draft:** Jiale Li.

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
