## [Decision Letter · Decision Letter 0]

17 Oct 2024

PONE-D-24-17979Lc - ms-based untargeted metabolomics reveals potential mechanisms of histologic chronic inflammation promoting prostate hyperplasiaPLOS ONE

Dear Dr. Li,

Thank you for submitting your manuscript to PLOS ONE. After careful consideration, we feel that it has merit but does not fully meet PLOS ONE’s publication criteria as it currently stands. Therefore, we invite you to submit a revised version of the manuscript that addresses the points raised during the review process. Please submit your revised manuscript by Dec 01 2024 11:59PM. If you will need more time than this to complete your revisions, please reply to this message or contact the journal office at plosone@plos.org. Please include the following items when submitting your revised manuscript:A rebuttal letter that responds to each point raised by the academic editor and reviewer(s). You should upload this letter as a separate file labeled 'Response to Reviewers'.A marked-up copy of your manuscript that highlights changes made to the original version. You should upload this as a separate file labeled 'Revised Manuscript with Track Changes'.An unmarked version of your revised paper without tracked changes. You should upload this as a separate file labeled 'Manuscript'.

We look forward to receiving your revised manuscript.

Kind regards,

Stanisław Jacek Wroński, M.D., Ph.D, FEBU

Academic Editor

PLOS ONE

Journal Requirements:

-https://doi.org/10.1016/j.cellsig.2020.109849

In your revision ensure you cite all your sources (including your own works), and quote or rephrase any duplicated text outside the methods section. Further consideration is dependent on these concerns being addressed.

4. Please remove your figures from within your manuscript file, leaving only the individual TIFF/EPS image files, uploaded separately. These will be automatically included in the reviewers’ PDF.

Additional Editor Comments:

Dear authors,

please find the following remarks:

There were substantial difficulties in finding suitable reviewers and obtaining approval from invited reviewers. Finally, the decision was made to send the manuscript back to the authors with a recommendation for significant revisions.

The authors should address the following issues prior to publication:

1.The authors should provide the full forms and significance of all abbreviations (e.g., PV) to enhance comprehension for general readers.

2.Were there any challenges related to ion suppression or enhancement during the LC-MS analysis, and what strategies were employed to address these effects?

3.Can the authors elaborate on the limitations encountered during the sample preparation process and how they were addressed to ensure the reliability of the LC-MS results?

By addressing these points, the manuscript will become more comprehensive and informative, thereby improving clarity and ensuring that the presented innovations are fully understood by the readership.

With compliments

Stanisław Wroński

Academic Editor

Reviewers' comments:

Reviewer's Responses to Questions

**Comments to the Author**

1. Is the manuscript technically sound, and do the data support the conclusions?

Reviewer #1: Yes

2. Has the statistical analysis been performed appropriately and rigorously? 

Reviewer #1: Yes

3. Have the authors made all data underlying the findings in their manuscript fully available?

Reviewer #1: Yes

4. Is the manuscript presented in an intelligible fashion and written in standard English?

Reviewer #1: Yes

5. Review Comments to the Author

Reviewer #1: This study reveals that histological chronic inflammation significantly alters prostate metabolic profiles, particularly affecting lipid metabolites such as free fatty acids, ceramides, sphingomyelins, and lysophosphatidylcholine. Utilizing LC-MS-based metabolomics, the research establishes these metabolic changes as key contributors to the pathogenesis of inflammation-induced prostate hyperplasia, offering new insights for prevention and treatment strategies. The manuscript is well-written, and I recommend its publication in PLOS ONE. However, I believe the authors should address the following issues prior to publication:

1.The authors should provide the full forms and significance of all abbreviations (e.g., PV) to enhance comprehension for general readers.

2.Were there any challenges related to ion suppression or enhancement during the LC-MS analysis, and what strategies were employed to address these effects?

3.Can the authors elaborate on the limitations encountered during the sample preparation process and how they were addressed to ensure the reliability of the LC-MS results?

By addressing these points, the manuscript will become more comprehensive and informative, thereby improving clarity and ensuring that the presented innovations are fully understood by the readership.

6. PLOS authors have the option to publish the peer review history of their article (what does this mean?). If published, this will include your full peer review and any attached files.

Reviewer #1: No

---

## [Author Response · Author response to Decision Letter 0]

24 Oct 2024

Response to reviewers

Dear editor and reviewers, 

Thank you for offering us an opportunity to improve the quality of our submitted manuscript " Lc - ms-based untargeted metabolomics reveals potential mechanisms of histologic chronic inflammation promoting prostate hyperplasia" (ID: PONE-D-24-17979R1). I received constructive and insightful comments. In this revision, we have addressed all of these suggestions. We hope the revised manuscript has now met the publication standard of your journal. We highlighted all the revisions in red colour. Below are our responses to the questions posed by the reviewers. 

Overall comment 1 

The authors should provide the full forms and significance of all abbreviations (e.g., PV) to enhance comprehension for general readers.

Response: Thank you for your suggestions. We have carefully and meticulously refined the full names of all abbreviations in this paper. We have red-flagged the changes in the article.

Comment 2 

Were there any challenges related to ion suppression or enhancement during the LC-MS analysis, and what strategies were employed to address these effects?

Response: Thank you very much for your professional question. Ion suppression or enhancement is a manifestation of matrix effect, which does exist during LC-MS analysis. The matrix effect in LC-MS is generally thought to occur because non-volatile components of the matrix compete with the substance to be measured, during ionization on the surface of the droplet, affecting the ionization efficiency at the electrospray interface. These non-volatile matrix components draw the droplets together, preventing them from cleaving into smaller microdroplets. The interfering substances that produce matrix effects fall into two categories. One group is called “endogenous substances”: substances that originate from the analyte itself and remain in the final extract, including salts, strongly polar compounds, surfactants, and lipids, amines, peptides, etc., that are structurally similar to the target compound. The other category is called “exogenous substances”: they do not come from the matrix itself, but from the external environment during the establishment of the method, including plastic and polymer residues, ion-pairing reagents, organic acids, buffer solutions, etc. In this experiment, I mainly take the substances from the matrix itself and retain them in the final extraction solution. In this experiment, I took four main programs to reduce the matrix effect. First, we adopted the better extraction scheme of liquid-liquid extraction for tissue metabolites as tested by want et al. The principle is that nonpolar compounds are more soluble in organic solvents than in water. On the contrary, ionic or polar compounds are more soluble in aqueous solutions than in organic solvents. Ultimately, the polar and nonpolar compounds will be extracted as much as possible. Second, after preexperimentation, we applied a variety of columns (HSS T3, BEH C8, BEH C18, CSH Phenyl-Hexyl) with different chromatographic separation conditions. It was found that the application of the columns and chromatographic conditions in this study had minimal matrix effects. Third, we used leucine enkephalin for mass calibration to reduce mass bias. Fourth, Progenesis QI (Waters, Nonlinear Dynamics, Newcastle, UK) is a better metabolomics data processing software, which has the functions of base; line filtering, peak alignment and deconvolution to reduce the experimental result error caused by matrix effect. Due to the fixed equipment, the replacement of ion source was not carried out in this experiment, and it was not possible to assess the extent of the matrix effect of most compounds on the ion source. These are the measures I took to reduce the matrix effect.

Comment 3 

Can the authors elaborate on the limitations encountered during the sample preparation process and how they were addressed to ensure the reliability of the LC-MS results?

Response: Thanks again for the question from the expert, there are indeed many limitations in the sample preparation process. The main ones are following items. First, sample contamination and sample stability. After the patient underwent prostate enlargement surgery, we collected the prostate tissue in sterile, enzyme-free cryopreservation tubes within 30 minutes and stored it in a -80°C refrigerator in the operating room. The vast majority of metabolites will not be degraded and transformed under ultra-low temperature conditions. Second, loss during sample processing. During sample processing, loss of target metabolites may occur due to the different extraction protocols for metabolites and the time required. Therefore, we performed pre-experimentation beforehand to choose a better extraction protocol and operated on dry ice throughout the process to reduce the loss of metabolites. Third, matrix effect. We ensured that all reagents and materials were of high quality and met the requirements for LC-MS analysis. Plastic devices that could interfere, such as bottles, funnels, beakers, etc., were avoided. Likewise the measures taken in the second expert question. Fourth, strict quality control. In the pre-treatment, the original samples were randomly disrupted between all the different groups to avoid batch effects between the comparison groups. For quality control (QC) and to improve the stability of the system, 10 μL of supernatant was transferred from each sample and mixed as QC, and 18.2 MΩ.cm pure water was used as a blank control group, and the QC samples and the blank control group would be injected at regular intervals (every 5 samples) to minimize the sample residues and to monitor the experimental stability. A blank solvent (Blank) was used to balance the instrument status and clean up the column residual material during the assay to avoid sample residue and contamination buildup problems. Preventive maintenance is achieved by pre-collecting QC mixed matrix samples to prevent situations such as unstable data.

[Author Name]: JiaLe Li

[Affiliation]: Clinical Medical College, North China University of Science and Technology 

[Mailing Address]: Tangshan, Hebei, China, 063210

[Phone Number]: +86 18352185307 

[Email]: 1654989167@qq.com

---

## [Decision Letter · Decision Letter 1]

14 Nov 2024

Lc - ms-based untargeted metabolomics reveals potential mechanisms of histologic chronic inflammation promoting prostate hyperplasia

PONE-D-24-17979R1

Dear Dr. Jiale Co

We’re pleased to inform you that your manuscript has been judged scientifically suitable for publication and will be formally accepted for publication once it meets all outstanding technical requirements.

Kind regards,

Stanisław Jacek Wroński, M.D., Ph.D, FEBU

Academic Editor

PLOS ONE

Dear Authors,

after careful consideration of revised version (code PONE-D-24-17979R1) and the reviewer's opinion, I conclude that in its present form the article can be approved for publication in POLS ONE.

The authors have effectively addressed all concerns raised, resulting in a significantly improved manuscript.

With compliments

Stanisław Wroński

Academic Editor

---

## [Editor Report · Acceptance letter]

15 Nov 2024

PONE-D-24-17979R1 

PLOS ONE

Dear Dr. Li, 

I'm pleased to inform you that your manuscript has been deemed suitable for publication in PLOS ONE. Congratulations! Your manuscript is now being handed over to our production team.

Kind regards, 

on behalf of

Dr. Stanisław Jacek Wroński 

Academic Editor

PLOS ONE